# Regional importation and asymmetric within-country spread of SARS-CoV-2 variants of concern in the Netherlands

**Alvin X Han[1]\*, Eva Kozanli[1], Jelle Koopsen[1], Harry Vennema[2], RIVM COVID-19 molecular epidemiology group[2], Karim Hajji[2], Annelies Kroneman[2], Ivo van Walle[2], Don Klinkenberg[2], Jacco Wallinga[2], Colin A Russell[1], Dirk Eggink[1,2], Chantal Reusken[2]\***

[1]Department of Medical Microbiology and Infection Prevention, Amsterdam University Medical Centers, Amsterdam, Netherlands; [2]Centre for Infectious Disease Control, National Institute for Public Health and the Environment, Bilthoven, Netherlands

**\*For correspondence:**
x.han@amsterdamumc.nl (AXH);
chantal.reusken@rivm.nl (CR)

**Group author details:**
RIVM COVID-19 molecular epidemiology group See page 11

## Abstract

**Background:** Variants of concern (VOCs) of SARS-CoV-2 have caused resurging waves of infections worldwide. In the Netherlands, the Alpha, Beta, Gamma, and Delta VOCs circulated widely between September 2020 and August 2021. We sought to elucidate how various control measures, including targeted flight restrictions, had impacted the introduction and spread of these VOCs in the Netherlands.

**Methods:** We performed phylogenetic analyses on 39,844 SARS-CoV-2 genomes collected under the Dutch national surveillance program.

**Results:** We found that all four VOCs were introduced before targeted flight restrictions were imposed on countries where the VOCs first emerged. Importantly, foreign introductions, predominantly from other European countries, continued during these restrictions. After their respective introductions into the Netherlands, the Alpha and Delta VOCs largely circulated within more populous regions of the country with international connections before asymmetric bidirectional transmissions occurred with the rest of the country and the VOC became the dominant circulating lineage.

**Conclusions:** Our findings show that flight restrictions had limited effectiveness in deterring VOC introductions due to the strength of regional land travel importation risks. As countries consider scaling down SARS-CoV-2 surveillance efforts in the post-crisis phase of the pandemic, our results highlight that robust surveillance in regions of early spread is important for providing timely information for variant detection and outbreak control.

**Funding:** None.

## Editor's evaluation

Han et al., analyze sequences from randomly sampled COVID-19 cases in the Netherlands to understand the impact of flight restrictions on the importation of SARS-CoV-2 variants. In line with prior observations and common wisdom, they find that targeted flight restrictions were not effective at preventing introductions of new lineages and that their early spread in the Netherlands was sustained by urban centers. These useful findings, based on unusually strong sequence collection techniques, can inform surveillance policy and improve basic understanding of the spread of SARS-CoV-2 variants.

## Introduction

Coronavirus disease 2019 (COVID-19) has resulted in excess morbidity and mortality across the world. In response, governments have implemented travel restrictions and nonpharmaceutical interventions in order to limit introductions and reduce transmission of severe acute respiratory syndrome coronavirus 2 (SARS-CoV-2; *Brauner et al., 2021*; *Chinazzi et al., 2020*; *Flaxman et al., 2020*). However, high levels of global infections have led to the evolution and emergence of variants of concern (VOCs) that are more transmissible, some of which encode putative mutations that evade immunity acquired from previous infection or vaccination (*Harvey et al., 2021*). These VOCs have led to resurging SARS-CoV-2 outbreaks, hampering efforts to contain and control the pandemic worldwide. Of note, four such VOCs arose into global prominence in late 2020, including Alpha (Nextclade 20I; PANGO lineage B.1.1.7), Beta (20 H; B.1.351), Gamma (20 J; P.1), and Delta (21 J; B.1.617.2), causing substantial levels of transmission worldwide, with Alpha and Delta being the most common VOCs globally in 2021 (*Campbell et al., 2021*).

The Alpha VOC was first reported in the United Kingdom (U.K.) during the fall of 2020 and found to be 43–90% more transmissible (*Davies et al., 2021a*; *Volz et al., 2021*) with greater mortality risks (*Davies et al., 2021b*; *Grint et al., 2021*) than previously existing variants. Of the 17 amino acids mutations found in Alpha, N501Y in the receptor-binding domain (RBD) of the spike protein was predicted to increase binding to the human angiotensin-converting enzyme 2 receptors (*Starr et al., 2020*). This is also a common mutation found in Beta (*Tegally et al., 2021*) and Gamma (*Faria et al., 2021*). On the other hand, the Delta VOC, first identified in India in October 2020 (*Cherian et al., 2021*), encodes P681R mutation in the furin cleavage site in spike protein and R203M mutation in the nucleocapsid protein that improves infectivity (*Syed et al., 2021*). Delta has also been linked to increased disease severity, as well as greater and longer viral shedding (*Ong et al., 2021*). In the U.K., where the VOC was first detected in April 2021, epidemiological modelling estimated the VOC to be 40–80% more transmissible than Alpha.

The four aforementioned VOCs were also introduced into the Netherlands, with the Alpha and Delta VOCs subsequently dominating infections in the country in 2021. In a bid to deter introductions and slow down the spread of VOCs, the Dutch government implemented targeted flight restrictions on countries where these VOCs had first emerged. Various non-pharmaceutical interventions were also implemented as the country experienced multiple waves of infections between 2020 and 2021. Since the end of 2020, the Dutch National Institute for Public Health and Environment scaled up its sequencing efforts under a random national surveillance program. This detailed surveillance program allows the monitoring of the introduction and spread of novel variants or specific mutations. Here, SARS-CoV-2 positive samples were randomly collected across the country and 39,844 high-quality SARS-CoV-2 whole genomes were sequenced between 22 September 2020 and 31 August 2021 (48 calendar weeks). We analyzed these sequences alongside epidemiological data to identify importation risks of novel variants into the Netherlands and characterize their subsequent patterns of spread within the country.

## Methods

### Whole genome sequencing

A total of 39,844 nasopharyngeal samples were randomly collected across all 25 GGD municipal health services across all municipalities in the Netherlands and were sequenced for whole SARS-CoV-2 genomes. These samples were collected through nationwide community testing programs including at test buses and health facilities (https://www.rivm.nl/coronavirus-covid-19/onderzoek/kiemsurveillance). Test-positive samples were randomly subsampled each week in each municipality to minimize case ascertainment bias. Only specimens with cycle threshold (Ct) values <30 were selected for whole genome sequencing.

As testing and samples were analyzed by 30–35 different laboratories across the country, different nucleic acid extraction methods were used (*Herrebrugh et al., 2021*). For samples analyzed by the laboratory of the Dutch National Institute for Public Health and Environment (38,260 samples; 96% of all samples analyzed), total nucleic acid was extracted using MagNApure 96 (MP96) with total nucleic acid kit small volume (Roche). RT-qPCR was performed on 5 µl total nucleic acid using TaqMan Fast Virus 1-Step Master Mix (Thermo Fisher) on Roche LC480 II thermal cycler with SARS-like beta

coronavirus (Sarbeco) specific E-gene primers and probe and EAV as described previously (*Corman et al., 2020*; *Scheltinga et al., 2005*).

Amplicon-based SARS-CoV-2 sequencing for was performed using the Nanopore protocol 'PCR tiling of COVID-19 virus (Version: PTC_9096_v109_revE_06FEB2020)' which is based on the ARTIC v3 amplicon sequencing protocol (*Artic Network, 2020*). Several modifications were made to the protocol as primer concentrations were increased from 0.125 to 1 pmol for the following amplicon primer pairs: in pool A amplicons 5, 9, 17, 23, 55, 67, 71, 91, 97 and in pool B amplicons 24, 26, 54, 64, 66, 70, 74, 86, 92, 98. Both libraries were generated using native barcode kits from Nanopore SQK-LSK109 (EXP-NBD104, EXP-NBD114 and EXP-NBD196) and sequencing was performed on a R9.4.1 flow cell multiplexing 2 up to 96 samples per sequence run. Consensus sequences (>50 x depth-of-coverage) are generated using an in-house bioinformatic pipeline developed by the Dutch National Institute for Public Health and Environment (https://github.com/RIVM-bioinformatics/SARS2seq/; *Zwagemaker, 2022*).

## Epidemiological data

All epidemiological data including the breakdown of positive cases by age group and weekly number of laboratory-confirmed cases in each Municipal and Regional Health Service region are provided by the Dutch National Institute for Public Health and Environment (https://www.rivm.nl/en/node/163991).

## Phylogenetic analyses

We performed ancestral reconstruction analyses for each VOC lineage to identify likely overseas introduction into the Netherlands at the continental level, differentiating the Netherlands from the rest of Europe. As proportions of cases for each VOC lineage are unknown for most countries, we subsampled global sequences downloaded from GISAID (https://www.gisaid.org; dataset up to 6 October 2021) by the proportion of COVID-19 cases reported per week for each country using data from the Johns Hopkins University, Center for Systems Science and Engineering (CSSE) (http://github.com/CSSEGISandData/COVID-19). We aimed to sample 100 sequences each week. To ensure that countries that are underreporting cases (*Gill et al., 2022*) were included in our analyses, at least one representative sequence was included for each country with reported cases that week. The actual number of sequences sampled each week for each variant may differ because of sequence availability and the requirement to have at least one sequence from each country each week. For sequences from the Netherlands, we also subsampled Dutch sequences by randomly subsampling the number of sequences in each GGD region each week to the corresponding relative number of reported cases in the same week for that GGD region. As there may be certain weeks when our aforementioned strategy sampled less non-Netherlands sequences than Netherlands sequences, we would resample a larger number of non-Netherlands sequences accordingly such that there are at least twice as many non-Netherlands sequences as Netherlands sequences each week. The subsampling procedure yielded 6,365 (2,369), 1,531 (90), 1,274 (102) and 6,929 (1,035) Alpha, Beta, Gamma, and Delta global (Dutch) sequences, respectively.

All sequences were aligned to hCoV-19/Wuhan/WIV04/2019 (WIV-04; EPI_ISL_402124) using MAFFT v7.427 (*Katoh and Standley, 2013*). Likely problematic sites (https://github.com/W-L/ProblematicSites_SARS-CoV2) along with untranslated regions in the 5' and 3' ends were masked. All maximum-likelihood (ML) phylogenetic trees were reconstructed using IQ-TREE (*Nguyen et al., 2015*) under the Hasegawa–Kishono–Yano nucleotide substitution model with a gamma-distributed rate variation among sites (HKY +G). We regressed the root-to-tip genetic distances against sampling dates using treetime v0.8.1 (*Sagulenko et al., 2018*) to assess the level of temporal signal, ensuring that none of the representative sequences were deemed molecular clock outliers.

Using these sequences, we then reconstructed approximate ML phylogenies using FastTree v2.1.11 (*Price et al., 2010*). All phylogenetically neighboring overseas sequences placed within two nodes away from any Dutch sequence were retained. This further reduced the number of sequences to a representative set of 3671, 496 and 575 and 2180 sequences for Alpha, Beta, Gamma and Delta VOCs respectively. We then reconstructed an ML phylogenetic tree again under the HKY +G nucleotide substitution model using IQ-TREE after removing any molecular clock outlying sequences identified by treetime. Here, however, we included the WIV-04 reference genome in the phylogeny reconstruction

which was used as an outgroup for tree rooting. We then time-scaled these ML phylogenies using treetime, which were then used as fixed tree topologies in BEAST v.1.10.4 (*Suchard et al., 2018*) to perform Bayesian discrete phylogeographical analyses at the continental level. Here, we performed 100 million MCMC generations, sampling every 1000 steps.

We randomly downsampled sets of Alpha (n=1389) and Delta (n=1342) variant sequences collected in the Netherlands. To understand within-country source-sink dynamics during early introductions and proliferation patterns during later periods, we used BEAST v.1.10.4 to perform continuous phylo-geographical analyses on these sequence data, using a relaxed random walk diffusion model and a Cauchy distribution model among branch heterogeneity in diffusion velocity (*Lemey et al., 2010*). We used HKY + G nucleotide substitution model and a Skygrid coalescent model (*Gill et al., 2013*; each grid point denoting one week) with a relaxed lognormal molecular clock. We inferred geographical coordinate input using the first four digits of postcodes (i.e. neighborhood level) associated with the sampled sequences. For sequences with identical postcodes, we randomly selected geographical coordinates corresponding to the postcode area using shapefiles provided by https://www.gadm.org. We performed two independent chains, each with 300 million MCMC generations for each variant analysis, sampling every 50,000 steps. The first 100 million steps were discarded as burn-in. Assessment of convergence (effective sample size >200) for each chain was performed using Tracer v1.71 (*Rambaut et al., 2018*). The posterior distribution of trees from both chains were combined and resampled at every 100,000 steps to generate the maximum clade credibility tree. Customized scripts from the SERAPHIM package (*Dellicour et al., 2016*) were used to extract the inferred phylogeographic reconstruction.

All tree visualizations were performed using baltic (https://github.com/evogytis/baltic; *Dudas, 2021*).

## Aggregated mobility data

We used publicly available mobility data from Google COVID-19 community mobility reports (https://www.google.com/covid19/mobility/) which contain daily anonymized location histories as a measure of people's movements. Google mobility data consisted of six categories that were measured relative to a baseline value. This baseline is the median mobility value between pre-pandemic weeks of 3 January and 6 February 2020. Categories include residence, parks, retail and recreation, groceries and pharmacies, working place and transit. Data for different regions of the Netherlands were available. We calculated aggregated nationwide mean mobility by averaging values across all regions for all categories except for residence and parks where the former has a reversed effect on relative mobility while the latter is affected by climate.

## Results

### SARS-CoV-2 infections and genotypes circulating in the Netherlands from September 2020 to August 2021

There were 1,792,759 laboratory-confirmed SARS-CoV-2 cases in the Netherlands during the study period between 22 September 2020 and 31 August 2021 (week 39/2020 to week 34/2021; *Figure 1A*). Similar to the first wave of the pandemic in the Netherlands in Spring 2020, most reported cases were attributed to the more densely populated regions of the country including North and South Holland, as well as North Brabant (*Figure 1B*) where the first local clusters of SARS-CoV-2 were also detected in March 2020 (*Oude Munnink et al., 2020*). 39,844 SARS-CoV-2 positive nasopharyngeal samples were randomly selected from 25 Municipal and Regional Health Service (GGD) regions across the Netherlands during this study period and sequenced to obtain whole virus genomes as part of the national SARS-CoV-2 genomic surveillance program.

Using NextClade lineage assignment (*Aksamentov et al., 2021*), the viruses sampled at the start of the study period were largely genotyped as clade 20 A and its daughter lineages, 20B and 20E (EU1) (*Figure 1B–C*). 20 A was the lineage that seeded the pandemic in Europe in March 2020. On the other hand, 20E (EU1) was first detected in Spain on June 2020 and spread widely across Europe due to the resumption of regional travel over summer 2020 (*Hodcroft et al., 2021*). Owing to rising case numbers, non-pharmaceutical interventions closing restaurants and nightlife establishments

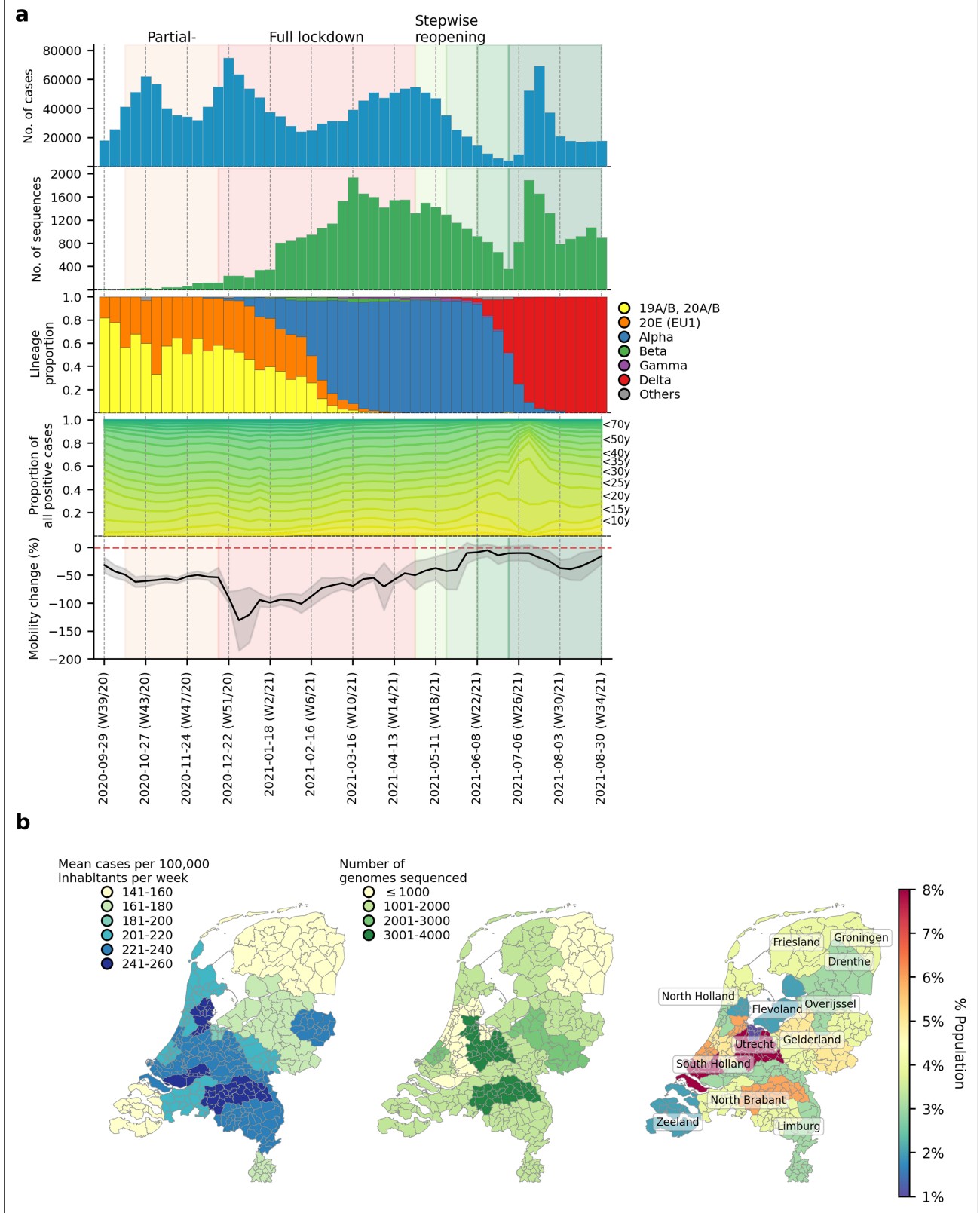

**Figure 1.** SARS-CoV-2 infections and VOCs circulating in the Netherlands from September 2020 to August 2021. (**A**) Weekly number of laboratory-confirmed SARS-CoV-2 cases (1st panel from the top) and sequenced genomes (2nd panel). Lineage proportions of sequences colored by NextClade genotype designations (3rd panel). Breakdown of positive cases by age group from data provided by the Dutch National Institute for Public Health and Environment (4th panel). Aggregated weekly average percentage change in mobility to the baseline in the Netherlands from Google's COVID-19

*Figure 1 continued on next page*

*Figure 1 continued*

community mobility reports. Baseline mobility is the median value from a 5-week period between 3 January 2020 and 6 February 2020, prior to the COVID-19 pandemic in Europe (5th panel). (**B**) Mean number of laboratory-confirmed cases per 100,000 inhabitants (data from the Dutch National Institute for Public Health and Environment; left panel); total number of sequenced genomes in different Municipal and Regional Health Service (GGD) regions over the entire study period (middle panel); Percentage of Dutch population residing in each GGD region (right panel).

were implemented on 14 October 2020. Cases dipped momentarily while both 20 A and 20E (EU1) remained co-circulating into December 2020.

The first Alpha sample was collected on 5 December 2020 in the national surveillance program prior to the full lockdown that closed all public venues, workplaces and schools on 15 December 2020. A curfew was also imposed later on 23 January 2021. A sharp drop in cases was observed after the implementation of the full lockdown. Alpha then displaced 20 A and 20E (EU1) over time to become the dominant circulating virus lineage by 16 February 2021 (week 6) for the rest of the lockdown period. Other VOCs such as Beta (N=422 sequences; first sequence was collected on 22 December 2020) and Gamma (N=350 sequences; first sequence was collected on 27 January 2021) were also detected by random surveillance in The Netherlands around the turn of the year but did not circulate to the levels of Alpha.

Alpha caused a rebound in cases around mid-March 2021, after which case numbers stabilized and eventually began to decline at the end of April 2021. The Dutch government began taking steps to relax restrictions around the same time, starting with the end of curfew and resumption of higher education during the week of 27 April 2021. The first Delta sample was collected in the previous week on 15 April 2021 and continued to accumulate in frequency. By week 25 (29 June 2021), the Delta VOC accounted for 24% of all weekly genomes sequenced.

Most restrictions were lifted in the same week by 26 June 2021. As SARS-CoV-2 prevalence declined over time, average nationwide mobility also increased steadily which eventually came close to pre-pandemic levels in June 2021 (mean percentage change relative to pre-pandemic baseline = –5.0% (s.d.=11.0%); *Figure 1A*). SARS-CoV-2 prevalence was at its lowest then with only 8,690 reported positive cases that week. Within only 1 week after reopening, however, weekly cases soared above 50,000 on weeks 26 and 27 (6–20 July 2021). With most infections attributed to Delta, the novel VOC replaced the Alpha VOC as the dominant lineage within the next 3 weeks as over 90% of randomly surveilled genomes were typed as Delta VOCs by mid-July.

Stratifying the number of weekly reported positive cases by patient age group, the relative proportions in case positive rates remained fairly consistent throughout the study period except for weeks 26 and 27 where the rapid increase in cases was largely attributed to individuals aged between 15 and 30 years (*Figure 1A*). One of the reasons behind widespread transmission among young adults then was super-spreading linked to nightlife venues (*Koopsen et al., 2022*). In response, the government shut nightlife establishments down again on 10 July 2021 (week 27). Case numbers fell promptly after but remained at over 30,000 new cases per week for the rest of the study period. The Delta VOC had in principle completely displaced Alpha by then with over 99% of randomly surveilled genomes sampled from August 2021 onwards.

## Overseas introduction of variants of concern

To deter the introductions of novel VOCs into the Netherlands, travel restrictions were imposed on countries where the VOCs first emerged, including the United Kingdom between December 2020 and March 2021 due to the emergence of Alpha; South Africa and Brazil between January and June 2021 due to Beta and Gamma respectively; and India from April to June 2021 due to Delta. These travel restrictions include a ban on all incoming passenger flights except for those carrying cargo and medical personnel, on top of an entry ban for all non-European Union residents (*Government of the Netherlands, 2021*). On the other hand, travel within parts of the Schengen Area in Europe, which includes the Netherlands, remained largely possible during this period. To identify likely where and when VOCs were actually introduced into the country, we subsampled a representative set of Dutch and overseas sequences collected over the same time period. We then reconstructed time-scaled, maximum likelihood (ML) phylogenies and used these fixed trees to perform discrete trait analyses using a Bayesian approach to infer likely overseas introductions at the continental level. This was done by identifying subtrees subtending Dutch sequences with ancestral states that were attributed to an

overseas origin (*Figure 2*). All four VOCs were already introduced into the Netherlands prior to the targeted flight restrictions that were imposed on countries where these VOCs first emerged.

Importantly, besides countries where travel restrictions were in place, we estimated multiple likely introduction events from other foreign countries into the Netherlands for all four VOCs (Alpha, n=100; Beta, n=7; Gamma, n=12; Delta, n=213). Given disparities in global sequencing efforts (*Brito et al., 2021*), the random surveillance strategy used in local sample collection, and low genetic diversity among SARS-CoV-2 genomes used to reconstruct ancestral states, we are unable to fully and reliably quantify the number of introductions attributed to different geographical regions. However, many of the estimated regions for these ancestral states were in Europe (Alpha, 71% of all estimated overseas introduction events; Beta, 29%; Gamma, 71%; Delta, 79%; *Figure 2*). Furthermore, these European introductions continue to occur during the targeted travel ban period. Inspecting the nearest phylogenetic ancestral taxon to the aforementioned subtrees, we found that many of these nearest overseas neighboring tips were detected in Belgium, Germany, France and Denmark where borders between the Netherlands remained open as well as other countries (e.g. Spain, U.K., Poland, U.S.) where no targeted travel restrictions were set in place (*Figure 2—figure supplement 1*). There was also no isolated period in time in which these VOCs were introduced into the Netherlands - introductions likely occurred repeatedly during the period when these VOCs were also proliferating within the country.

## Within-country transmission dynamics of the Alpha and Delta VOCs

To further elucidate the transmission dynamics of the Alpha and Delta VOCs within the Netherlands, we performed continuous phylogeographic analyses using separate downsampled sets of Alpha and Delta sequence data (*Figure 3*). For the first four weeks since the initial detection of both VOCs within the country, introductions and phylogenetic branch movements were mostly concentrated in the more populous regions of the country, including North and South Holland, Utrecht and North-Brabant, forming a core of early dominant locations. During this period, dispersal events to regions outside of these GGD regions occurred as well but are relatively less frequent. However, as local infections were seeded in these areas, bidirectional exchanges in phylogenetic branches between different regions emerged throughout the country. These bidirectional exchanges continued to increase as prevalence of the VOC grew over time, even amidst a strict lockdown in the case of Alpha (*Figure 4*). In particular for the Delta VOC, we observed a rapid spike in inter-regional spread upon the week of 22–28 June 2021, with >400% estimated increase in total phylogenetic branch movements by 6 July 2021 (week 26). This significant rise in inter-regional exportation events likely contributed to the soaring case numbers observed between weeks 25 and 27 (22 June – 13 July 2021).

## Discussion

Even if international travel restrictions are in place, the Netherlands is still highly vulnerable to importation risks of novel SARS-CoV-2 variants from its regional neighbors due to border policies within the European Union. As such, this regional vulnerability is not unique to the Netherlands and has been reported in other European countries as well (*Lemey et al., 2021*; *Michaelsen et al., 2021*; *Osnes et al., 2021*). Importantly, regional introductions of novel lineages often drive new waves of infections in Europe (*Lemey et al., 2021*). Prior to September 2020, the dominant variant lineages (i.e. 20 A and 20E (EU1)) that circulated the Netherlands were already seeded by imports from its European neighbours (*Hodcroft et al., 2021*; *Nadeau et al., 2021*; *Oude Munnink et al., 2020*). In fact, the initial introduction of SARS-CoV-2 in the Netherlands in February 2020 were attributed to travelers who visited Northern Italy where the earliest sustained European SARS-CoV-2 transmission network was seeded (*Oude Munnink et al., 2020*; *Worobey et al., 2020*). Here, we showed that all four VOC lineages detected in the country up to August 2021 also originated mainly from its European neighbors. Importantly, regional importation risks persisted throughout the period these VOCs circulated the country and overlapped with periods where targeted flight restrictions were imposed on countries where these VOCs first emerged. The emergence of the Omicron VOC in southern Africa in November 2021 (*Viana et al., 2022*) again led to reactionary targeted flight restrictions by several countries in the Global North, including the Netherlands which was still amidst a surging Delta infection wave. However, the Omicron VOC was already detected in samples collected one week before the imposed

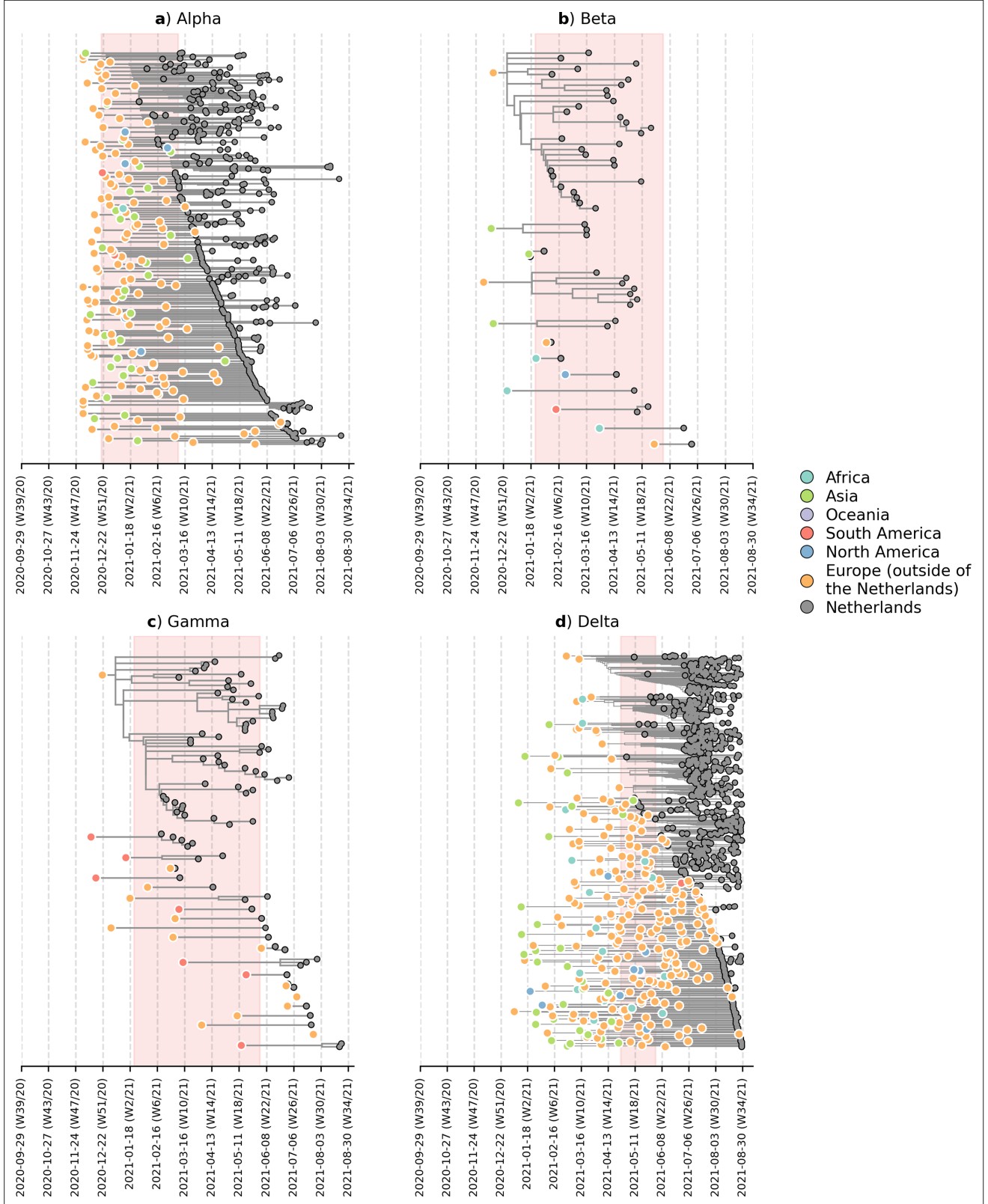

**Figure 2.** Likely overseas introduction of VOC lineages into the Netherlands at the continental level. For each VOC lineage, a time-scaled maximum likelihood phylogeny using the Dutch and their nearest overseas neighboring sequences was inferred. Discrete trait analyses were performed to infer the likely continental region of ancestral states. Subtrees or singletons with ancestral nodes attributed to an overseas origin but subtend only Dutch

*Figure 2 continued on next page*

*Figure 2 continued*

sequences are drawn. Shaded plot area denotes the timespan when a targeted flight restriction was imposed on the country where the VOC lineage first emerged (i.e. (**A**) Alpha, United Kingdom.; (**B**) Beta, South Africa; (**C**) Gamma, Brazil; (**D**) Delta, India).

The online version of this article includes the following figure supplement(s) for figure 2:

**Figure supplement 1.** Distribution of countries of the nearest overseas neighboring taxon to Dutch subtrees.

**Figure supplement 2.** Likely overseas introduction of VOC lineages into the Netherlands at the continental level (based on phylogenetic analyses using an independently subsampled set of sequence data from that used in *Figure 2*).

**Figure supplement 3.** Distribution of countries of the nearest overseas neighboring taxon to Dutch subtrees.

flight restriction and did not prevent it to rapidly become the dominant VOC circulating in the Netherlands by the end of 2021 (https://www.rivm.nl/coronavirus-covid-19/virus/varianten/omikronvariant). Previous studies showed that travel restrictions are only useful if restrictions barred arrivals from most countries provided that local incidence is low in the first place (*Russell et al., 2021*).

Due to disparities in global genomic surveillance efforts and the lack of travel history information among the sampled Dutch individuals, we could not make more precise and accurate phylogeographic inferences on overseas introduction of VOCs into the Netherlands (*Lemey et al., 2020*). Furthermore, there were countries that are underrepresented or missing in our subsampling of non-Netherlands sequences. While we ensured that at least one sequence from each country with confirmed cases and genomic data was included in our analyses (see Methods), there may be overseas introductions that we could not account for. We repeated our overseas phylogeographic analyses with an independent bootstrap subsample of sequence data (*Figure 2—figure supplements 2–3*). There are differences in the estimated distributions of countries with nearest overseas neighboring sequences to Dutch subtrees (*Figure 2—figure supplements 1 and 3*). However, our conclusions that there were multiple introductions from other European countries before and after travel restrictions were imposed on

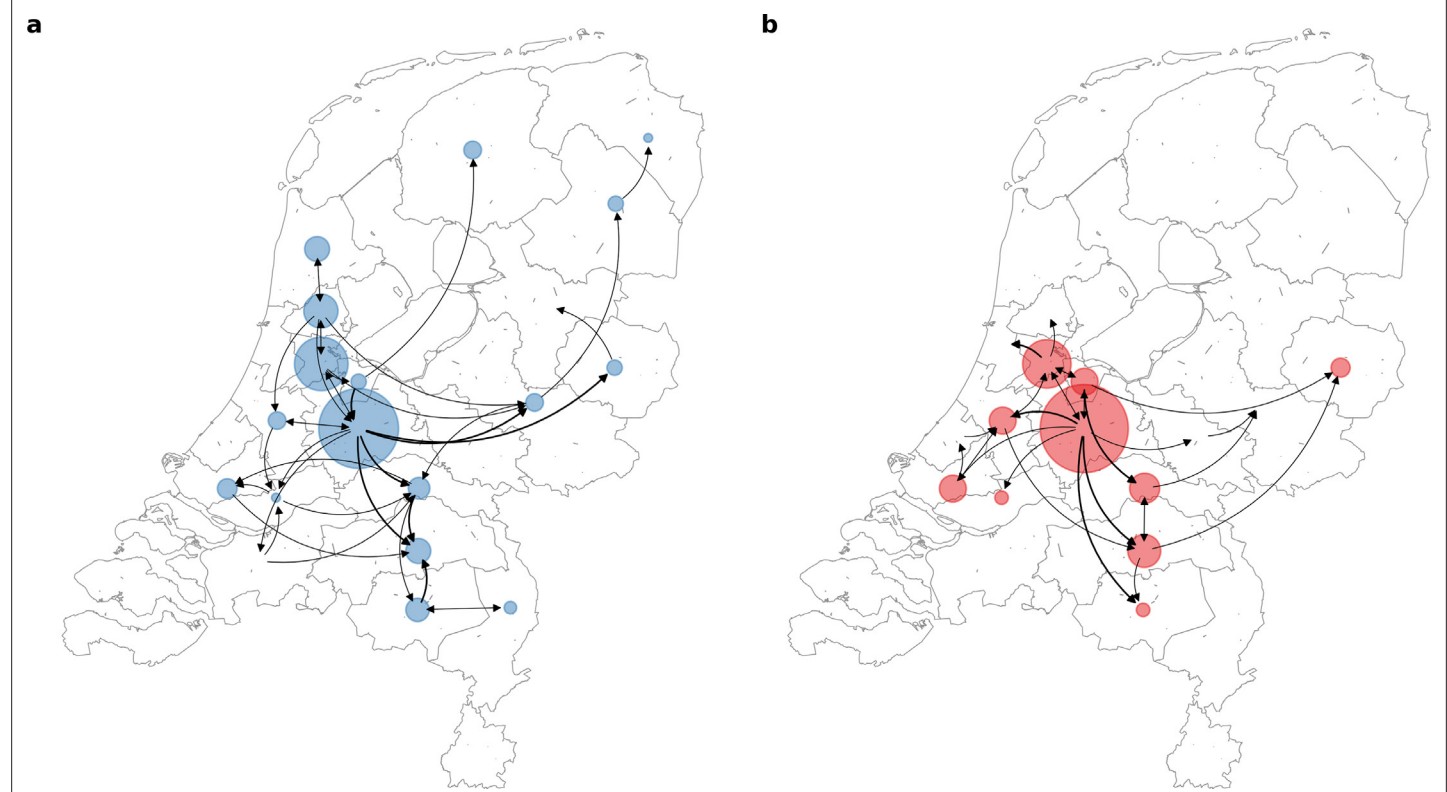

**Figure 3.** Source-sink dynamics the Alpha and Delta variants of concern (VOC) within the Netherlands during the first four weeks after their respective detection. (**A**) Alpha (between 2 and 29 December 2020); (**B**) Delta (between 20 April and 18 May 2021).

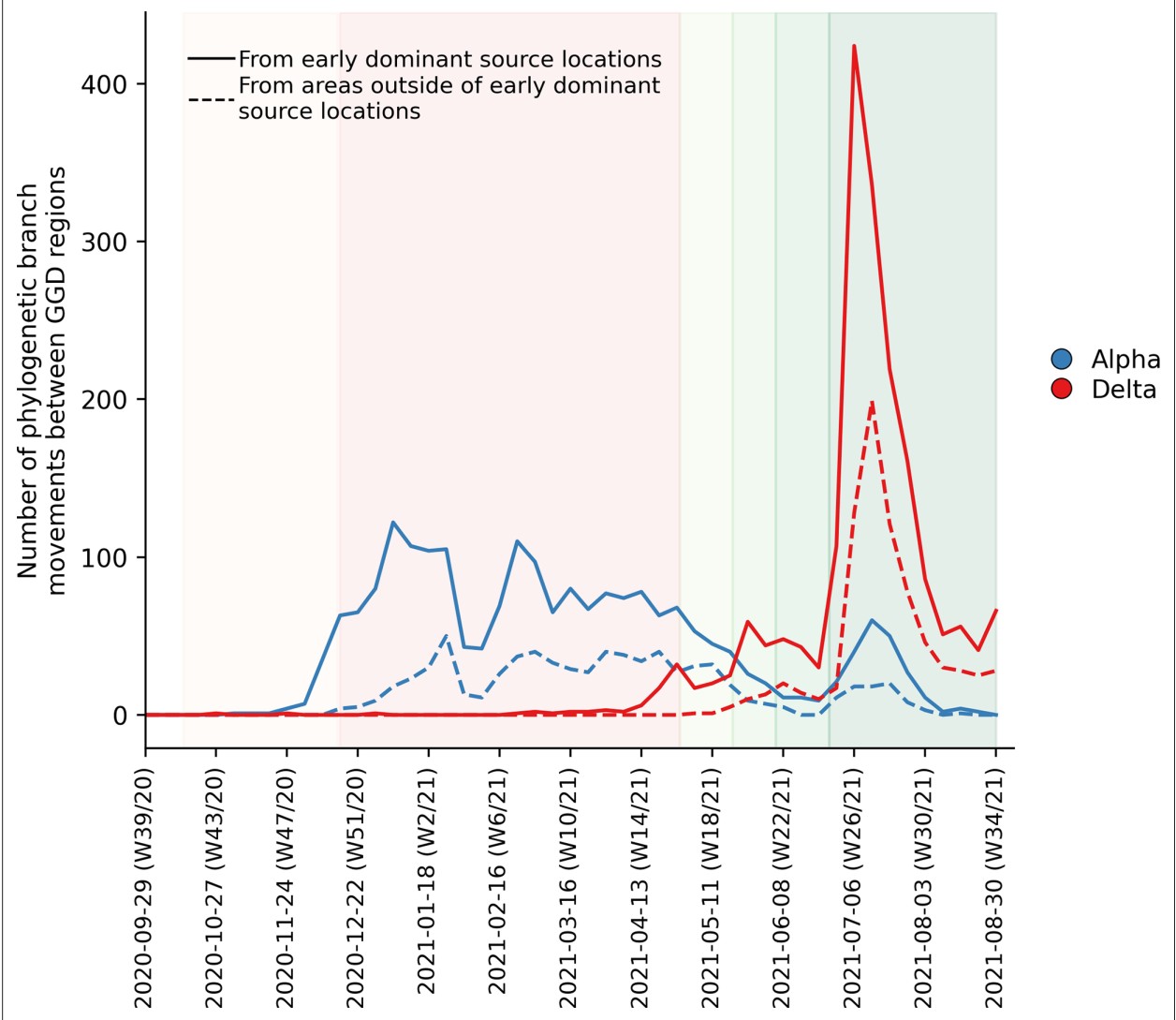

**Figure 4.** Estimated number of phylogenetic branch movements and growth rate estimations of Alpha and Delta VOCs in the Netherlands. Solid line shows the number of branch movements from early dominant source regions including North and South Holland, Utrecht and Brabant. Dashed line shows number of branch movements from areas outside of these early dominant source locations.

countries where VOCs first emerged remain evident in the bootstrap analysis (*Figure 2—figure supplement 2*).

We also found that early introductions of VOCs, specifically the Alpha and Delta VOCs, are more likely found in populous regions of the Netherlands, including Utrecht, North, and South Holland where larger cities are located that are also international and regional travel hubs. These areas constitute a core cluster of dominant source locations that also exported infections to the rest of the country during first few weeks after the VOC's introduction into the Netherlands. As the number of infections in areas outside of these dominant source locations increase over time, bidirectional exchanges would also become more frequent. This type of asymmetric spatial spread dynamics had been previously shown in the U.K. as well and was found to enhance the intrinsic transmissibility of Alpha (*Kraemer et al., 2021*). Additionally, enhanced mobility has also been previously linked to the resurgence of outbreaks across Europe (*Hodcroft et al., 2021*; *Lemey et al., 2021*). Recent work also showed that increased mobility and population mixing drove the rapid dissemination of Delta in the U.K. (*McCrone et al., 2021*). While our analyses do not provide a causal relationship between the relaxation of non-pharmaceutical interventions and frequency of export events, the asymmetric exportation frequencies from dominant source locations, increased human mobility in weeks 25–27 across the country as well

as the intrinsic higher transmissibility of Delta relative to Alpha likely all contributed to the widespread spike in cases in the Netherlands.

Novel and fitter variants of SARS-CoV-2 will likely continue to emerge in the future. Our results, along with others, show that unless well-coordinated actions are taken across Europe to mitigate importation risks (*Ruktanonchai et al., 2020*), targeted travel restrictions implemented by individual European countries will neither prevent nor slow down the introduction of novel variants. Our work also shows that early within-country spread of VOCs may be taken into future consideration in future genomic surveillance strategies, especially as countries are gradually considering scaling down SARS-CoV-2 surveillance efforts. Both the Alpha and Delta VOCs were first detected in the early dominant source locations, usually those that are more populous with greater international connections, and circulated mostly within these areas during the initial period after introduction. As such, a robust level of surveillance efforts should still be maintained in these dominant source locations to provide timely actionable information on novel variant detection as well as infection control. These surveillance efforts should encompass a minimal level of clinical diagnostic testing capacity be maintained to ensure clinical genomic surveillance remains sensitive enough for early detection of novel variants (*Han et al., 2022*). Wastewater surveillance could also be included to facilitate early variant detection and identify cryptic transmissions amid falling testing rates (*Karthikeyan et al., 2022*).

## Data availability

All sequencing data have been deposited in the GISAID database (https://www.gisaid.org) under the accession codes listed in . All codes for our analyses are available at https://github.com/AMC-LAEB/nl_sars-cov-2_genomic_epi_2022; *Han, 2022*.

## Acknowledgements

We thank the administrators of the GISAID database for supporting rapid and transparent sharing of genomic data during the COVID-19 pandemic and all our colleagues sharing data on GISAID. A full list acknowledging the authors submitting genome sequence data used in this study can be found in *Supplementary file 2*. AXH and CAR were supported by ERC NaviFlu (no. 818353). C.A.R. was also supported by NIH R01 (5R01AI132362-04) and an NWO Vici Award (09150182010027).

## Additional information

Group author details

**RIVM COVID-19 molecular epidemiology group**

**Lynn Aarts**: Centre for Infectious Disease Control, National Institute for Public Health and the Environment, Bilthoven, Netherlands; **Sanne Bos**: Centre for Infectious Disease Control, National Institute for Public Health and the Environment, Bilthoven, Netherlands; **Annemarie van den Brandt**: Centre for Infectious Disease Control, National Institute for Public Health and the Environment, Bilthoven, Netherlands; **Sharon van den Brink**: Centre for Infectious Disease Control, National Institute for Public Health and the Environment, Bilthoven, Netherlands; **Jeroen Cremer**: Centre for Infectious Disease Control, National Institute for Public Health and the Environment, Bilthoven, Netherlands; **Kim Freriks**: Centre for Infectious Disease Control, National Institute for Public Health and the Environment, Bilthoven, Netherlands; **Ryanne Jaarsma**: Centre for Infectious Disease Control, National Institute for Public Health and the Environment, Bilthoven, Netherlands; **Dennis Schmitz**: Centre for Infectious Disease Control, National Institute for Public Health and the Environment, Bilthoven, Netherlands; **Euníce Then**: Centre for Infectious Disease Control, National Institute for Public Health and the Environment, Bilthoven, Netherlands; **Bas van der Veer**: Centre for Infectious Disease Control, National Institute for Public Health and the Environment, Bilthoven, Netherlands; **Lisa Wijsman**: Centre for Infectious Disease Control, National Institute for Public Health and the Environment, Bilthoven,

Netherlands; **Florian Zwagemaker**: Centre for Infectious Disease Control, National Institute for Public Health and the Environment, Bilthoven, Netherlands

## Competing interests

RIVM COVID-19 molecular epidemiology group: The other authors declare that no competing interests exist.

## Funding

| Funder | Grant reference number | Author |
|---|---|---|
| European Research Council | ERC NaviFlu (no. 818353) | Alvin X Han |
| National Institutes of Health | 5R01AI132362-04 | Colin A Russell |
| Nederlandse Organisatie voor Wetenschappelijk Onderzoek | 09150182010027 | Colin A Russell |

The funders had no role in study design, data collection and interpretation, or the decision to submit the work for publication.

## Author contributions

Alvin X Han, Conceptualization, Formal analysis, Supervision, Validation, Investigation, Visualization, Methodology, Writing – original draft, Project administration, Writing – review and editing; Eva Kozanli, Jelle Koopsen, Formal analysis, Validation, Investigation, Visualization, Methodology, Writing – original draft, Writing – review and editing; Harry Vennema, Don Klinkenberg, Jacco Wallinga, Resources, Data curation, Methodology, Writing – review and editing; RIVM COVID-19 molecular epidemiology group, Data curation, Formal analysis, Investigation, Methodology, Resources, Software, Validation, Writing – review and editing; Karim Hajji, Resources, Data curation, Software, Methodology; Annelies Kroneman, Ivo van Walle, Resources, Data curation, Methodology; Colin A Russell, Conceptualization, Supervision, Funding acquisition, Project administration, Writing – review and editing; Dirk Eggink, Chantal Reusken, Conceptualization, Resources, Data curation, Supervision, Funding acquisition, Methodology, Project administration, Writing – review and editing

## Author ORCIDs

Alvin X Han (iD) http://orcid.org/0000-0001-6281-8498

## Ethics

Human subjects: The Centre for Clinical Expertise at the National Institute for Public Health and the Environment (RIVM) assessed the research proposal following the specific conditions as stated in the law for medical research involving human subjects. The work described was exempted for further approval by the ethical research committee. Pathogen surveillance is a legal task of the RIVM and is carried out under the responsibility of the Dutch Minister of Health, Welfare and Sports. The Public Health Act (Wet Publieke Gezondheid) provides that RIVM may receive pseudonymized data for this task without informed consent. All necessary patient/participant consent has been obtained and the appropriate institutional forms have been archived, and any patient/participant/sample identifiers included were not known to anyone (e.g., hospital staff, patients or participants themselves) outside the research group so cannot be used to identify individuals.

## Decision letter and Author response

Decision letter https://doi.org/10.7554/eLife.78770.sa1
Author response https://doi.org/10.7554/eLife.78770.sa2

# Additional files

## Supplementary files

• Supplementary file 1. Accession codes of all Dutch sequencing data that have been deposited in the GISAID database (https://www.gisaid.org).

• Supplementary file 2. Acknowledgement table of authors of global genome sequence data deposited in the GISAID database that were used in this study.

• MDAR checklist

• Reporting standard 1. SRQR Checklist.

## Data availability

All sequencing data have been deposited in the GISAID database (https://www.gisaid.org) under the accession codes listed in Supplementary File 1. All codes for our analyses are available at https://github.com/AMC-LAEB/nl_sars-cov-2_genomic_epi_2022, copy archived at swh:1:rev:0a3c0a3e7d3959587e82e743162e28b45fa42dd7.

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
