## [Editor Report]

Han et al., analyze sequences from randomly sampled COVID-19 cases in the Netherlands to understand the impact of flight restrictions on the importation of SARS-CoV-2 variants. In line with prior observations and common wisdom, they find that targeted flight restrictions were not effective at preventing introductions of new lineages and that their early spread in the Netherlands was sustained by urban centers. These useful findings, based on unusually strong sequence collection techniques, can inform surveillance policy and improve basic understanding of the spread of SARS-CoV-2 variants.

---

## [Decision Letter]

**Decision letter after peer review:**

Thank you for submitting your article "Regional importation and asymmetric within-country spread of SARS-CoV-2 variants of concern in the Netherlands" for consideration by *eLife*. Your article has been reviewed by 3 peer reviewers, one of whom is a member of our Board of Reviewing Editors, and the evaluation has been overseen by Miles Davenport as the Senior Editor. The reviewers have opted to remain anonymous.

Essential revisions:

1) Please streamline the narrative arc and logic of the paper. The reviewers questioned (including in the consultation session) what the estimation of variant growth rates adds to the analysis. It seems important to emphasize that the qualitatively estimated impacts of flight bans are if anything underestimated here, given other concurrent NPIs; please clarify which restrictions were in place. Please also see Reviewer 2's comment on the role of nightclubs.

2) There were technical concerns about the soundness of the results. It appears as though only one chain was used; several are traditional and safer. Biases might be introduced by the use of a strict vs. relaxed clock. A discrete phylogeographic model would be more appropriate. These improvements would lend greater confidence to the results. There should also be some treatment (at least in the Discussion) of the potential impact of sampling biases (inside and outside The Netherlands) on the results.

3) The reviewers found many of the figures confusing or unhelpful. Please revise them in light of the suggestions below.

*Reviewer #1 (Recommendations for the authors):*

1. I realize this is pretty foundational to phylogeographic methods (and welcome being referred to a textbook or classic paper I have missed), but how sensitive are the results – e.g., estimates of importation from ground travel/various areas---to sampling biases? For this question, I'm mostly thinking about deliberate subsampling biases (e.g., 2:1 Netherlands to non-Netherlands).

2. I'm curious about the potential magnitude of these effects due to biases in case ascertainment that might be poorly understood. In the U.S., there were many-fold differences in populations' access to testing and probability of being reported as a case, conditional on infection. These differences had strong socioeconomic and spatial associations. It seems deeply misleading to me to describe the sequencing as "random." It's not, unless case identification was random, which it wasn't. Maybe you could say something like "sequences were collected from a random sample of cases." But it would be good in the Methods and maybe the main text to describe exactly what is known about case ascertainment biases in The Netherlands over the study period, and the potential impacts of case ascertainment biases within The Netherlands and between countries on the estimated flows.

3. There's one mention of flight bans but otherwise "restrictions" is used. It would be useful to describe the restriction/ban on air travel precisely for people less familiar with the policies.

4. The github repo needs better documentation and structure to ensure reproducibility.

*Reviewer #2 (Recommendations for the authors):*

I have found the manuscript in need of several improvements:

1. The focus of the manuscript is somewhat diffuse and at times misguided, where for example the analysis of SARS-CoV-2 introductions into the Netherlands seems fine, others are questionable like using continuous instead of discrete phylogeography, and the analysis of VOC growth rate differences feeling a bit pointless, especially in light of the narrative the authors seem to push regarding the role of nightclubs in SARS-CoV-2 transmission. The authors should rethink the structure of the manuscript and what they want to say with it. If it's finding that travel restrictions do not achieve their goals – perfect! But adding analyses to beef up the manuscript afterward doesn't add much to the story.

2. There are a couple of technical issues that should be addressed. Firstly, all Bayesian MCMC analyses describe a single chain which is not standard. Ideally, at least two independent runs are expected for any MCMC analysis. Secondly, some of the figures don't seem like they convey much information, e.g. Figure 1C, Figure 3, and the bottom panel in Figure 4, and their use should therefore be reconsidered.

Overall, the manuscript has potential but requires substantial effort to improve, starting with its scope.

As stated in the public review part I think the authors should reframe, and refocus their manuscript. I think a study on the ineffectiveness of targeted flight cancellations is worthwhile but I don't see the added value (or good justification) of using continuous (instead of discrete) phylogeography or the logistic regressions.

I should also say that the narrative being pushed about nightclub reopenings is very flimsy and potentially confounded with the Netherlands having an age-tiered vaccination programme with age groups being given access around June 2021. This is not mentioned at all in the text and offers an alternative hypothesis for the distribution of cases across age groups during the reopening. I understand the authors are officially not claiming that there is strong evidence for it but when it comes up more than once in the text and is part of the discussion I personally would like it to be backed up with more evidence and all caveats addressed properly.

Line 76 "The four aforementioned VOCs also emerged in the Netherlands" – 'emerged' is often understood as 'originated', better to say 'arrived to' or 'dominated' here.

Figure 1A – Middle panel should say "Lineage proportions" for clarity, consider labelling each dominant area with the lineage that it represents since the legend in Figure 1C contains too many colours.

Figure 1A – The bottom panel should say "Mobility change (percent)" to make it intuitive.

Figure 1B – I have a personal preference for neatly-delineated colour maps and it's up to the authors if they want to change the limits between incidence and sequences so they're, for example, multiples of 20s and 500s, respectively.

Figure 1C – Not sure the tree is adding much here.

Line 143 "and continued to accumulate in frequencies" – change to 'frequency'.

Figure 2 – Worth shading European states with good sequencing programmes with a similar hue? Are those countries not flying passengers in?

Figure 3 – This figure is so messy perhaps it's not worth using? If the authors are dead-set on having some representation of migrations maybe there are better ways of doing it rather than a mess that says "there's a lot of movement"?

Figure 4 – Panel B is not labelled.

Geographic diffusion analyses indicate that more populous regions in the Netherlands play a large role in virus introduction and dissemination but surely this is entirely expected, given that such areas host transportation hubs for introductions and have more hosts to seed other areas.

Line 278 "locatthed" -> "located".

Paragraph 2 of the discussion contains passages that are more suitable for the Results section.

A very large dataset is analysed in BEAST so why did the authors decide to not use a relaxed clock and infer the rate naively? I find it very hard to believe that a dataset spanning close to a year would not have a temporal signal to inform the rate.

In all MCMC analyses, only one MCMC chain is mentioned. I'm sure the authors are aware that most studies are expected to run at least two independent ones, particularly with complex data like these.

*Reviewer #3 (Recommendations for the authors):*

1. Line 53: "Coronavirus^-1^9 disease" should be "coronavirus disease 2019".

2. Line 88: samples were randomly selected for sequencing, not genomes.

3. Variants of concern (VOC), variants, and genotypes are used interchangeably. It would help to just pick of these and use them consistently.

4. I have a few suggestions regarding the visualization of Figure 1:

4.1. I would suggest using the Pangolin lineages as labels in panel C, and only list variants that are the focus of the study (all other lineages can be listed as "other").

4.2. The colors for 20D, 20I, 21A, 21I, and 21J are very similar (same in Figure 4). I would suggest using more distinctive colors to make it easier to see which color represents each of the variants.

4.3. It would be helpful to indicate the first detection of Α, Β, Γ, and Δ in Figure 1a.

5. Line 129. Did S Gene Target Failure (SGTF) influence the first detection of Α?

6. Figure 3. Panels B and D are hard to read, and may not be necessary to include. The advantage of only showing panels A and C is that their size can be increased.

7. Discussion. It would be helpful to start the discussion with a summary of the main findings. As raised before, I am not sure if the effect of international travel restrictions can really be distinguished from other interventions. Moreover, as Omicron was not part of the current study, I would refrain from speculating on the impact of those targeted flight restrictions as it is outside of the scope of the current study.

8. Limitations of the study are missing in the discussion.

9. Line 278: "locathed" should be "located".

10. Line 310. Methods lack a description of the used nucleic acid extraction method and diagnostic assay. Methods should also include a description of the used bioinformatics pipeline with specifics for how consensus genomes were generated (e.g. minimum depth and minimum frequency threshold to call consensus).

---

## [Author Response]

Reviewer #1 (Recommendations for the authors):1. I realize this is pretty foundational to phylogeographic methods (and welcome being referred to a textbook or classic paper I have missed), but how sensitive are the results – e.g., estimates of importation from ground travel/various areas---to sampling biases? For this question, I'm mostly thinking about deliberate subsampling biases (e.g., 2:1 Netherlands to non-Netherlands).

To clarify, we did not subsample at a 2:1 non-Netherlands to Netherlands sequence ratio for our global/Netherlands phylogenetic analyses. Instead, the number of sequences sampled per country per week is proportional to the number of reported cases that week. We ensured that we have at least one sequence from each country in each week. We aimed to sample around 100 sequences each week (the number may differ because of available sequences and the requirement to have at least one sequence from each country each week). As our sampling strategy may result in less non-Netherlands sequences than Netherlands sequences for some VOCs each week, we would then resample the non-Netherlands sequences again with a larger sample size, ensuring that we have at least twice as many non-Netherlands as Netherlands sequence that week.

Overall, we aim to be as comprehensive as possible to have all countries represented each week in our sampling strategy. We have now made this clearer in our methods:

Line 464: “As proportions of cases for each VOC lineage are unknown for most countries, we subsampled global sequences downloaded from GISAID (https://www.gisaid.org; dataset up to 6 October 2021) by the proportion of COVID-19 cases reported per week for each country using data from the Johns Hopkins University, Center for Systems Science and Engineering (CSSE) (http://github.com/CSSEGISandData/COVID-19). We aimed to sample 100 sequences each week. To ensure that countries that are underreporting cases (Gill et al., 2022) were included in our analyses, at least one representative sequence was included for each country with reported cases that week. The actual number of sequences sampled each week for each variant may differ because of sequence availability and the requirement to have at least one sequence from each country each week. We also subsampled Dutch sequences based on the weekly number of cases in different GGD regions as described above. As there may be certain weeks when our aforementioned strategy sampled less non-Netherlands sequences than Netherlands sequences, we would resample a larger number of non-Netherlands sequences accordingly such that there are at least twice as many non-Netherlands sequences as Netherlands sequences each week. The subsampling procedure yielded 6,365 (2,369), 1,531 (90), 1,274 (102) and 6,929 (1,035) Α, Β, Γ and Δ global (Dutch) sequences respectively.”

To check if our results are robust to sampling, we repeated our analyses with an independent sample bootstrap and found similar results as before:

Line 360: “We repeated our overseas phylogeographic analyses with an independent bootstrap subsample of sequence data (Figure 2 —figure supplements 2-3). There are differences in the estimated distributions of countries with nearest overseas neighboring sequences to Dutch subtrees (Figure 2 —figure supplement 1 and 3). However, our conclusions that there were multiple introductions from other European countries before and after travel restrictions were imposed on countries where VOCs first emerged remain evident in the bootstrap analysis (Figure 2 —figure supplement 2).”

2. I'm curious about the potential magnitude of these effects due to biases in case ascertainment that might be poorly understood. In the U.S., there were many-fold differences in populations' access to testing and probability of being reported as a case, conditional on infection. These differences had strong socioeconomic and spatial associations. It seems deeply misleading to me to describe the sequencing as "random." It's not, unless case identification was random, which it wasn't. Maybe you could say something like "sequences were collected from a random sample of cases." But it would be good in the Methods and maybe the main text to describe exactly what is known about case ascertainment biases in The Netherlands over the study period, and the potential impacts of case ascertainment biases within The Netherlands and between countries on the estimated flows.

We have now included further details in the Methods on the sources of the samples and how they were sampled for analyses:

Line 426: “These samples were collected through nationwide community testing programs including at test buses and health facilities (https://www.rivm.nl/coronavirus-covid-19/onderzoek/kiemsurveillance). Test-positive samples were randomly subsampled each week in each municipality to minimize case ascertainment bias. Only specimens with cycle threshold (Ct) values < 30 were selected for whole genome sequencing.”

3. There's one mention of flight bans but otherwise "restrictions" is used. It would be useful to describe the restriction/ban on air travel precisely for people less familiar with the policies.

We have now included more details on the travel restriction imposed on countries where VOCs first emerged:

Line 227: “To deter the introductions of novel VOCs into the Netherlands, travel restrictions were imposed on countries where the VOCs first emerged, including the United Kingdom between December 2020 and March 2021 due to the emergence of Α; South Africa and Brazil between January and June 2021 due to Β and Γ respectively; and India from April to June 2021 due to Δ. These travel restrictions include a ban on all incoming passenger flights except for those carrying cargo and medical personnel, on top of an entry ban for all non-European Union residents (Government of the Netherlands, 2021). On the other hand, travel within the Schengen Area of the European Union, which includes the Netherlands, remained possible during this period.”

4. The github repo needs better documentation and structure to ensure reproducibility.

We have now provided more detailed documentation in the GitHub repository to reproduce our results and figures.

Reviewer #2 (Recommendations for the authors):I have found the manuscript in need of several improvements:1. The focus of the manuscript is somewhat diffuse and at times misguided, where for example the analysis of SARS-CoV-2 introductions into the Netherlands seems fine, others are questionable like using continuous instead of discrete phylogeography, and the analysis of VOC growth rate differences feeling a bit pointless, especially in light of the narrative the authors seem to push regarding the role of nightclubs in SARS-CoV-2 transmission. The authors should rethink the structure of the manuscript and what they want to say with it. If it's finding that travel restrictions do not achieve their goals – perfect! But adding analyses to beef up the manuscript afterward doesn't add much to the story.

As stated in the Introduction, this work aims to identify where novel SARS-CoV-2 VOCs are likely introduced into the Netherlands and then characterize the spread of these VOCs in the country after their introduction. We selected different methods for our phylogeographic analyses based on the resolution of data available. We opted to perform discrete phylogeography to identify where likely overseas importation events were occurring from since only country/regional-level of geographic data was available to us. On the other hand, as postcodes of sampled Dutch individuals were available to us and since within-country spread can be assumed to occur mostly in a geographically homogenous way across the country, we employed continuous phylogeography to characterize the within-country spread of the novel variants. Continuous phylogeography methods have also been used to characterize within-country SARS-CoV-2 spread dynamics in several countries, including the United Kingdom (Kraemer et al., Science 2021; McCrone et al., medRxiv 2022), Rwanda (Butera et al., Nat Comm 2021) and Belgium (Bollen et al., Nat Sci Rep 2021).

We have removed the logistic regression analyses on VOC growth rate differences as we agree it adds little to the narrative.

2. There are a couple of technical issues that should be addressed. Firstly, all Bayesian MCMC analyses describe a single chain which is not standard. Ideally, at least two independent runs are expected for any MCMC analysis.

We had performed two independent Bayesian MCMC chains for our BEAST analyses. The maximum clade credibility trees were generated by combining the posterior distribution of trees from both chains. This is now explicitly stated in the Methods:

Line 592: “We performed two independent chains, each with 300 million MCMC generations for each variant analysis, sampling every 50,000 steps. The first 100 million steps were discarded as burn-in. Assessment of convergence (effective sample size > 200) for each chain was performed using Tracer v1.71 (Rambaut et al., 2018). The posterior distribution of trees from both chains were combined and resampled at every 100,000 steps to generate the maximum clade credibility tree.”

Secondly, some of the figures don't seem like they convey much information, e.g. Figure 1C, Figure 3, and the bottom panel in Figure 4, and their use should therefore be reconsidered.

We have removed Figure 1C and the bottom panel for Figure 4. We have revised and simplified Figure 3 to show the early within-country source-sink dynamics of Α and Δ during the first four weeks after their initial detection in the Netherlands.

Overall, the manuscript has potential but requires substantial effort to improve, starting with its scope.As stated in the public review part I think the authors should reframe, and refocus their manuscript. I think a study on the ineffectiveness of targeted flight cancellations is worthwhile but I don't see the added value (or good justification) of using continuous (instead of discrete) phylogeography or the logistic regressions.

See response to comment 1 above.

I should also say that the narrative being pushed about nightclub reopenings is very flimsy and potentially confounded with the Netherlands having an age-tiered vaccination programme with age groups being given access around June 2021. This is not mentioned at all in the text and offers an alternative hypothesis for the distribution of cases across age groups during the reopening. I understand the authors are officially not claiming that there is strong evidence for it but when it comes up more than once in the text and is part of the discussion I personally would like it to be backed up with more evidence and all caveats addressed properly.

We agree and have removed all sections suggestive of nightlife reopening as the main driving force behind the spike in Δ transmissions in the Netherlands during July 2021.

Line 76 "The four aforementioned VOCs also emerged in the Netherlands" – 'emerged' is often understood as 'originated', better to say 'arrived to' or 'dominated' here.

We have now rephrased to: “The four aforementioned VOCs were also introduced into the Netherlands…”.

Figure 1A – Middle panel should say "Lineage proportions" for clarity, consider labelling each dominant area with the lineage that it represents since the legend in Figure 1C contains too many colours.

We have changed the y-label of the middle panel to “*Lineage proportions*”. We have also simplified

Figure 1A – The bottom panel should say "Mobility change (percent)" to make it intuitive.

The y-label is now “Mobility change (%)”.

Figure 1B – I have a personal preference for neatly-delineated colour maps and it's up to the authors if they want to change the limits between incidence and sequences so they're, for example, multiples of 20s and 500s, respectively.

We have now changed the bins to multiples of 20s or 1000s.

Figure 1C – Not sure the tree is adding much here.

As mentioned above, Figure 1C is now removed.

Line 143 "and continued to accumulate in frequencies" – change to 'frequency'.

Corrected.

Figure 2 – Worth shading European states with good sequencing programmes with a similar hue? Are those countries not flying passengers in?

We could not further delineate between European countries with different sequencing intensity in Figure 2 as the discrete phylogeographic analyses were performed at the continental level. However, we have now provided this stratification in the figure supplements 1 and 3 to Figure 2 for the nearest neighbouring overseas sequences to Dutch sequences. Flights were possible between most European countries during the study period albeit with varying vaccination and/or quarantine requirements.

Figure 3 – This figure is so messy perhaps it's not worth using? If the authors are dead-set on having some representation of migrations maybe there are better ways of doing it rather than a mess that says "there's a lot of movement"?

We have now revised Figure 3 to make it more informative to support our findings.

Figure 4 – Panel B is not labelled.

Figure 4B has been removed.

Geographic diffusion analyses indicate that more populous regions in the Netherlands play a large role in virus introduction and dissemination but surely this is entirely expected, given that such areas host transportation hubs for introductions and have more hosts to seed other areas.

We agree and that is why we performed the within-country continuous phylogeographic analyses to confirm our null expectations.

Line 278 "locatthed" -> "located".

Corrected.

Paragraph 2 of the discussion contains passages that are more suitable for the Results section.

We have now shifted those passages into the Results section, specifically:

Line 180: “As SARS-CoV-2 prevalence declined over time, average nationwide mobility also increased steadily which eventually came close to pre-pandemic levels in June 2021 (mean percentage change relative to pre-pandemic baseline = -5.0% (s.d. = 11.0%); Figure 1A).”

A very large dataset is analysed in BEAST so why did the authors decide to not use a relaxed clock and infer the rate naively? I find it very hard to believe that a dataset spanning close to a year would not have a temporal signal to inform the rate.

We redid our BEAST analyses using a relaxed lognormal clock.

Line 583: “We used HKY+G nucleotide substitution model and a Skygrid coalescent model (Hill and Baele, 2019) (each grid point denoting one week) with a relaxed lognormal molecular clock.”

In all MCMC analyses, only one MCMC chain is mentioned. I'm sure the authors are aware that most studies are expected to run at least two independent ones, particularly with complex data like these.

See response to comment 2 above.

Reviewer #3 (Recommendations for the authors):1. Line 53: "Coronavirus^-1^9 disease" should be "coronavirus disease 2019".

Corrected.

2. Line 88: samples were randomly selected for sequencing, not genomes.

Corrected.

3. Variants of concern (VOC), variants, and genotypes are used interchangeably. It would help to just pick of these and use them consistently.

We have now solely used “variants of concern (VOC)” throughout the text.

4. I have a few suggestions regarding the visualization of Figure 1:4.1. I would suggest using the Pangolin lineages as labels in panel C, and only list variants that are the focus of the study (all other lineages can be listed as "other").

In response to comment 3 by reviewer 2, we have now removed Figure 1C.

4.2. The colors for 20D, 20I, 21A, 21I, and 21J are very similar (same in Figure 4). I would suggest using more distinctive colors to make it easier to see which color represents each of the variants.

We have now simplified the figure by distinguishing only the pre-Α and VOC lineages by different colours.

4.3. It would be helpful to indicate the first detection of Α, Β, Γ, and Δ in Figure 1a.

This is now included in Figure 1A.

5. Line 129. Did S Gene Target Failure (SGTF) influence the first detection of Α?

No.

6. Figure 3. Panels B and D are hard to read, and may not be necessary to include. The advantage of only showing panels A and C is that their size can be increased.

In response to comment 3 by reviewer 2, we have revised Figure 3 to make it more informative to support our findings.

7. Discussion. It would be helpful to start the discussion with a summary of the main findings. As raised before, I am not sure if the effect of international travel restrictions can really be distinguished from other interventions. Moreover, as Omicron was not part of the current study, I would refrain from speculating on the impact of those targeted flight restrictions as it is outside of the scope of the current study.

See response to comment 1 above.

8. Limitations of the study are missing in the discussion.

We have now expanded the Discussion on the limitations with our study:

Line 354: “Due to disparities in global genomic surveillance efforts and the lack of travel history information among the sampled Dutch individuals, we could not make more precise and accurate phylogeographic inferences on overseas introduction of VOCs into the Netherlands (Lemey et al., 2020). Furthermore, there were countries that are underrepresented or missing in our subsampling of non-Netherlands sequences. While we ensured that at least one sequence from each country with confirmed cases and genomic data was included in our analyses (see Methods), there may be overseas introductions that we could not account for. We repeated our overseas phylogeographic analyses with an independent bootstrap subsample of sequence data (Figure 2 —figure supplements 2-3). There are differences in the estimated distributions of countries with nearest overseas neighboring sequences to Dutch subtrees (Figure 2 —figure supplement 1 and 3). However, our conclusions that there were multiple introductions from other European countries before and after travel restrictions were imposed on countries where VOCs first emerged remain evident in the bootstrap analysis (Figure 2 —figure supplement 2).”

9. Line 278: "locathed" should be "located".

Corrected.

10. Line 310. Methods lack a description of the used nucleic acid extraction method and diagnostic assay. Methods should also include a description of the used bioinformatics pipeline with specifics for how consensus genomes were generated (e.g. minimum depth and minimum frequency threshold to call consensus).

We now provide the required information in the Methods:

Line 432: “As testing and samples were analyzed by 30-35 different laboratories across the country, different nucleic acid extraction methods were used (Herrebrugh et al., 2021). For samples analyzed by the laboratory of the Dutch National Institute for Public Health and Environment (38,260 samples; 96% of all samples analyzed), total nucleic acid was extracted using MagNApure 96 (MP96) with total nucleic acid kit small volume (Roche). RT-qPCR was performed on 5μl total nucleic acid using TaqMan Fast Virus 1-Step Master Mix (Thermo Fisher) on Roche LC480 II thermal cycler with SARS-like β coronavirus (Sarbeco) specific E-gene primers and probe and EAV as described previously (Corman et al., 2020; Scheltinga et al., 2005).

Amplicon-based SARS-CoV-2 sequencing for was performed using the Nanopore protocol “PCR tiling of COVID-19 virus (Version: PTC_9096_v109_revE_06FEB2020)” which is based on the ARTIC v3 amplicon sequencing protocol (Artic Network, 2020). Several modifications were made to the protocol as primer concentrations were increased from 0.125 to 1 pmol for the following amplicon primer pairs: in pool A amplicons 5, 9, 17, 23, 55, 67, 71, 91, 97 and in pool B amplicons 24, 26, 54, 64, 66, 70, 74, 86, 92, 98. Both libraries were generated using native barcode kits from Nanopore SQK-LSK109 (EXP-NBD104, EXP-NBD114 and EXP-NBD196) and sequencing was performed on a R9.4.1 flow cell multiplexing 2 up to 96 samples per sequence run. Consensus sequences (>50x depth-of-coverage) are generated using an in-house bioinformatic pipeline developed by the Dutch National Institute for Public Health and Environment (https://github.com/RIVM-bioinformatics/SARS2seq/).”